# Prognostic effect of pretreatment albumin-to-alkaline phosphatase ratio in human cancers: A meta-analysis

Xiaoli Guo[1], Qijiu Zou[1], Jiaxin Yan[1], Xingxing Zhen[2], Hongmei Gu[1]*

1 Department of Radiology, Affiliated Hospital of Nantong University, Nantong University, Nantong, Jiangsu, China, 2 Department of Radiology, Nantong Tumor Hospital, Nantong University, Nantong, Jiangsu, China

☯ These authors contributed equally to this work.

* guhongmei71@163.com

**Data Availability Statement:** All relevant data are within the paper and its Supporting Information files.

**Funding:** The authors received no specific funding for this work.

## Abstract

### Purpose

It has been demonstrated that, for various types of cancer, the pretreatment albumin/alkaline phosphatase ratio (AAPR) was a prognostic factor. Therefore, in order to determine AAPR's prognostic effect on cancer, the meta-analysis was hereby performed.

### Patients and methods

The relevant studies conducted before November 10, 2019, were comprehensively searched in Web of Science, PubMed, and Embase. HRs(hazard ratios) with related 95% CIs(confidence intervals) were adopted to estimate AAPR's prognostic impact on overall survival (OS) & disease-free survival (DFS).

### Results

Our meta-analysis involved thirteen cohort studies, which included 5,204 cases of 8 types. The results of this meta-analysis indicated that higher AAPR was corrected with better OS (pooled HR = 0.52; 95%CI = 0.47–0.58; $P<0.001$) and DFS (pooled HR = 0.55; 95%CI = 0.47–0.66; $P<0.001$). Subgroup analysis on OS was based on the cancer system, treatment methods, and cutoff value. Moreover, higher AAPR was statistically in associated with lighter infiltration (pooled OR = 0.79; 95%CI = 0.73–0.85; $P<0.001$), no lymph nodes metastasis (pooled OR = 0.89; 95%CI = 0.83–0.95; $P = 0.001$), and no distant metastasis (pooled OR = 0.92; 95%CI = 0.86–0.99; $P = 0.028$).

### Conclusion

Higher AAPR was related to better prognosis of cancer, and in cancer therapy, AAPR could be taken as a promising marker of prognosis. It might help physicians to select the most appropriate treatments by evaluating the current status of patients with cancer. Future multi-center prospective clinical trials were required to verify its applications.

**Competing interests:** The authors have declared that no competing interests exist.

## Introduction

As defined by WHO, cancer is a severe health problem worldwide. Its rate of occurrence is still increasing due to genetic mutations, environmental pollution, growth, aging of the population, and other various risks. It was estimated that there were 1,762,450 new cancer cases and 606,880 deaths in the United States in 2019 [1]. Nowadays, numerous prognostic markers, including microRNAs, long non-coding RNAs, and genes [2, 3], have been established to predict the survival time of cancer patients, most of which are either expensive or hard to be obtained in routine clinical practice. Hence, it is essential for us to identify a new prognostic marker, which should be cheap and easily obtained in a standardized manner.

Cancer is a complex disorder that may affect various parts of the human body and affect the metabolism systems, such as bone/muscle system and patients' nutrition conditions [4]. Previous studies demonstrated that tumor-related nutritional assumption and immune responses were correlated to tumor progression and development [5]. Recently, a variety of serum pretreatment markers have been evaluated and taken to assess whether they could provide valuable prognosis information in patients with cancer, including neutrophil-to-lymphocyte ratio (NLR), C-reactive protein to albumin ratio (CAR), and prognostic nutritional index (PNI) [6–8]. The rate of albumin to alkaline phosphatase (AAPR) was the ratio of serum albumin(ALB) level to alkaline phosphatase(ALP) level. In 2015, Anthony et al. first formed a novel inflammation-based marker—AAPR through integrating ALB and ALP, and demonstrated that the lower AAPR predict inferior overall survival (OS) in patients with hepatocellular carcinoma (HCC) compared with higher AAPR [9]. In recent years, more evidence has shown that AAPR could be used to predict the outcomes/survival of patients with various cancer, including lung cancer [10–13], nasopharyngeal carcinoma [14, 15], hepatocellular carcinoma [9, 16, 17], renal cell carcinoma [18], cholangiocarcinoma [19], upper tract urothelial carcinoma [20], and breast cancer [21]. However, considering the inevitable heterogeneity of various studies, the prognostic impact of AAPR has not been comprehensively investigated. Therefore, the meta-analysis was used for assessing AAPR's association with the clinical outcomes of cancer.

## Methods

### Search strategy

The protocol for this study was registered on PROSPERO (CRD42020163017). The meta-analysis was performed according to the statement in PRISMA (S1 Table) [22]. The following databases were comprehensively searched, including Embase, PubMed, and Web of Science (before November 10, 2019), where the following terms were used as the keywords: "albumin/alkaline phosphatase ratio," "albumin to alkaline phosphatase," "albumin to alkaline phosphatase ratio" or "AAPR," and "tumor, cancer, malignancy, neoplasms or carcinoma." Detailed search strategies were shown in S2 Table.

### Criteria for inclusion & exclusion

In these studies, the inclusion criteria were adopted as follows: (1) articles investigated the association with AAPR and cancer prognosis; (2) HRs(hazard ratios) and their 95%CIs(confidence intervals) for the prognosis were provided; (3) the cutoff value of AAPR had been reported; (4) the text was prepared in English.

While the following exclusion criteria were adopted: (1) reviews, comments, case reports or letters; (2) studies with insufficient data for HRs and 95%CIs; (3) duplicated studies; (4) animal studies; (5) no full text in English.

## Data extraction & quality assessment

These studies were separately assessed by Zou QJ and Zheng XX. The differences were resolved through discussions with a third investigator Yan JX, who collected the following data: (1) Basic information: the first author, country, publication date, study period, disease type, number of patients, the cutoff value for AAPR, and treatment method. (2) Clinicopathological features: gender, age, tumor invasion, distant metastasis, lymph node metastasis, and histological grade. (3) Prognostic data: follow-up time and survival outcomes, HRs and their 95%CIs for OS and DFS in univariate/multivariate analyses. The multivariate results could balance other factors, so they were extracted preferentially.

The quality of selected papers was evaluated according to the Newcastle-Ottawa Scale (NOS), which could be divided into three parts: selection, outcome, and comparability [23]. Each study could get a NOS score (0–9), where a review with a score of ≥6 could be regarded as a high-quality study. Detailed information about the NOS score were shown in S3 Table.

## Statistical analysis

The AAPR's prognostic effect on disease-free survival (DFS)/overall survival (OS) in cancer patients was assessed based on pooled HRs with related 95%CIs. In these studies, Higgins I-squared ($I^2$) statistic, together with Cochran's Q, was used to assess heterogeneity. A random effects model was adopted when statistical heterogeneity was found ($I^2 > 50\%$, $P < 0.1$); in other cases, the fixed effects model should be used. HR<1 (higher AAPR used as reference) showed a lower risk of worse outcomes for higher AAPR; meanwhile, if $P < 0.05$ and 95%CI<1, it would be deemed as statistically significant. Pooled ORs (odds ratios) and related 95%CI were used to estimate the relationship between AAPR and clinical features, including lymph node metastasis, infiltration, and distant metastasis. Egger's test, Begg's test, and funnel plot analysis were used to assess publication bias; funnel plot asymmetry ($P < 0.05$) illustrated that there might be a significant publication bias [24, 25], in which case, the trim/fill method of Duval and Tweede was used to evaluate publication bias's potential effect [26]. For evaluating the stability of the results, a sensitivity analysis was performed through precluding individual studies sequentially. All these statistical analyses were performed according to Stata version 15.1(StataCorp, College Station, TX).

# Results

## Literature retrieval and study characteristics

A flow diagram regarding literature retrieval was expressed in Fig 1. A total of 448 potential articles were collected through retrieving Web of Science, PubMed, and Embase. Among these studies, 105 articles were removed due to duplication. After the titles and abstracts had been screened by two investigators, 324 studies were excluded. Subsequently, 19 full-text articles were reviewed for eligibility. Finally, this meta-analysis involved 13 studies in total (5,204 patients).

For the 13 literature, the features were presented in Table 1. All these studies were retrospective ones published from 2015 to 2019. Two special studies were conducted based on two and three cohorts, respectively [9, 17]. Twelve studies were from China, and one study was from Korea, which investigated eight different types of cancers: small cell lung cancer(SCLC), hepatocellular carcinoma(HCC), cholangiocarcinoma(CCA), non-small cell lung cancer (NSCLC), nasopharyngeal carcinoma(NPC), breast cancer(BC), upper tract urothelial carcinoma(UTUC), and renal cell carcinoma(RCC). Thirteen studies reported the relationship

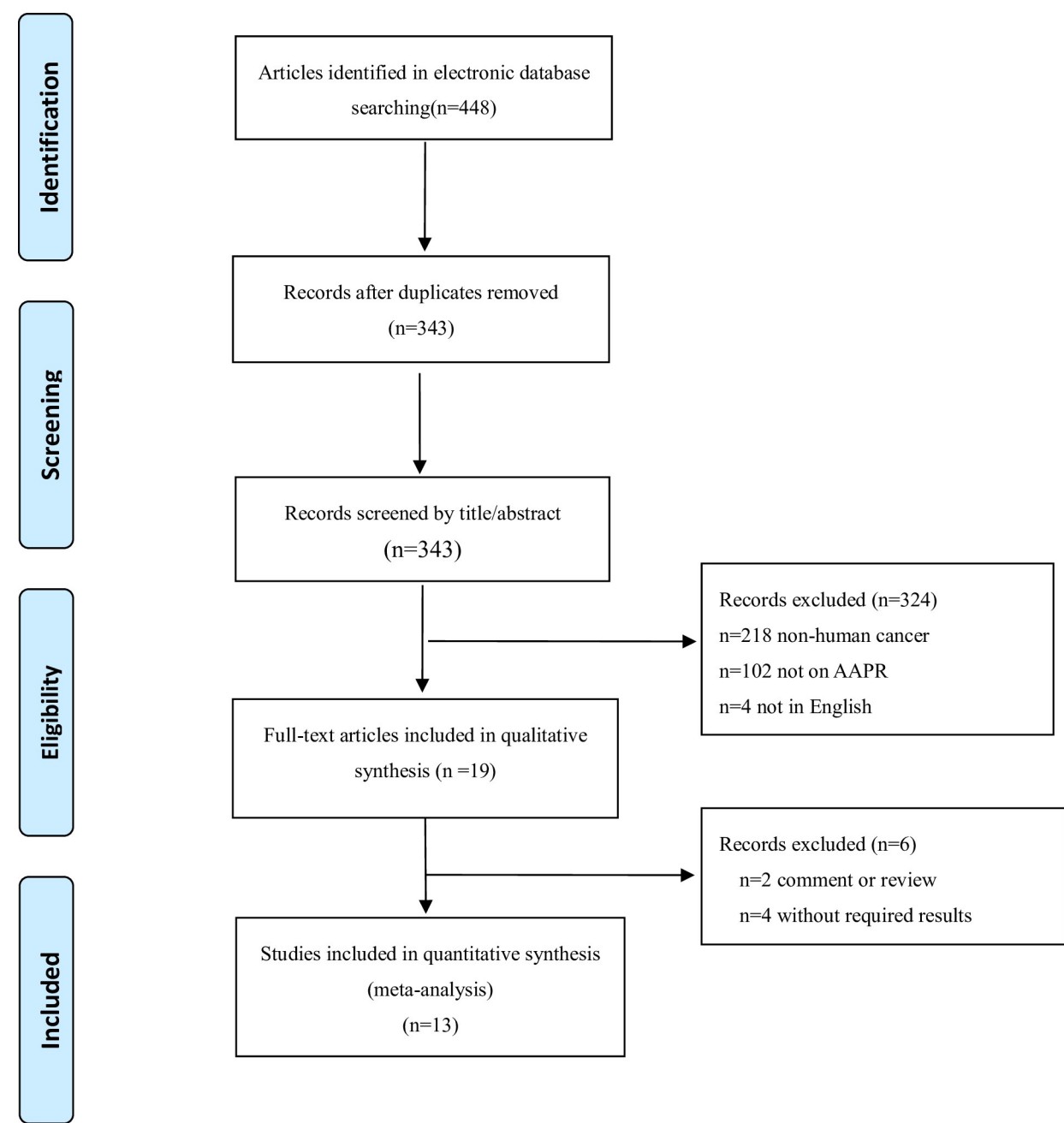

**Fig 1. A flow diagram for the literature assessment process.**

between AAPR and OS, while only three studies presented the association between AAPR and DFS. In all selected studies, NOS scores varied from 7 to 9, with a median of 8.

## Meta-analysis on OS

A number of 5,204 patients were included from 13 studies for the analysis of pooled HR for OS. Since no obvious heterogeneity was in these studies ($I^2$ = 3.3%, $P$ = 0.416), HR and 95%CI were combined for analysis using the fixed effects model. According to the pooled analysis, it

**Table 1. The characteristics of the included studies.**

| Author | Year | Country | Disease type | Treatment | Follow-up (month) | Cut off | Study period | Patients(n) | Survival | NOS score |
|---|---|---|---|---|---|---|---|---|---|---|
| Li SJ *et al.* | 2019 | China | NSCLC | Surgery | 60 | 0.57 | 2013–2015 | 390 | 0S/DFS | 9 |
| Li D *et al.* | 2019 | China | NSCLC | Mixed | NA | 0.36 | 2007–2013 | 290 | OS | 8 |
| Li XG *et al.* | 2019 | China | SCLC | Radiotherapy | NA | 0.61 | 2013–2015 | 122 | OS | 7 |
| Zhang *et al.* | 2019 | China | NSCLC | Surgery | Median47 (2–96) | 0.64 | 2006–2010 | 567 | 0S/DFS | 9 |
| Nie *et al.* | 2017 | China | NPC | Chemotherapy | Median16.6 (1–66.6) | 0.447 | 2008–2011 | 209 | OS | 9 |
| Kim *et al.* | 2019 | Korea | NPC | radiotherapy | Median50.6 (NA) | 0.4876 | 1996–2016 | 100 | OS | 8 |
| Chan *et al.* (training) | 2015 | China | HCC | Surgery | NA | 0.68 | 2007–2011 | 217 | 0S/DFS | 8 |
| Chan *et al.* (validation) | 2015 | China | HCC | Surgery | Median38.9 (0.1–95.4) | 0.68 | 2007–2011 | 256 | 0S/DFS | 9 |
| Cai *et al.* | 2018 | China | HCC | Palliative treatments | NA | 0.38 | 2006–2010 | 237 | OS | 7 |
| Chen *et al.* (training) | 2018 | China | HCC | TACE | NA | 0.439 | 2009–2013 | 372 | OS | 8 |
| Chen *et al.* (validation1) | 2018 | China | HCC | Palliative treatments | NA | 0.439 | 2009–2014 | 202 | OS | 8 |
| Chen *et al.* (validation2) | 2018 | China | HCC | TACE | NA | 0.439 | 2013–2014 | 82 | OS | 7 |
| Xiong *et al.* | 2019 | China | CCA | Surgery | Median21 (NA) | 0.41 | 2002–2014 | 303 | OS | 9 |
| Long *et al.* | 2019 | China | BC | Surgery | NA | 0.525 | 2011–2013 | 746 | OS | 8 |
| Tan *et al.* | 2018 | China | UTUC | Surgery | 60 | 0.58 | 2003–2016 | 692 | OS | 9 |
| Xia *et al.* | 2019 | China | RCC | Surgery | Median 50(30.4–83) | 0.39 | 2004–2014 | 419 | OS | 9 |

**Abbreviations:** NSCLC, non-small-cell lung cancer; SCLC, small-cell lung cancer; NPC, nasopharyngeal carcinoma; HCC, hepatocellular carcinoma; CCA, cholangiocarcinoma; BC, breast cancer; UTUC, upper tract urothelial carcinoma; RCC, renal cell carcinoma; OS, overall survival; DFS, disease-free survival; NOS: Newcastle-Ottawa Scale; NA: not available

was concluded that higher AAPR would lead to better OS in cancer patients (pooled HR = 0.52; 95%CI = 0.47–0.58; *P*<0.001, Fig 2). Furthermore, subgroup analyses of the included studies were conducted based on cancer system, treatment method, and cutoff value (Table 2), which concluded that higher AAPR would lead to a longer OS for patients with respiratory cancer (pooled HR = 0.58; 95%CI = 0.48–0.70; *P*<0.001), urinary cancer (pooled HR = 0.52; 95%CI = 0.40–0.68; *P*<0.001), digestive cancer (pooled HR = 0.51; 95%CI = 0.43–0.60; *P*<0.001), and other cancers (pooled HR = 0.40; 95%CI = 0.26–0.62; *P*<0.001). For the subgroup involving the treatment method, it was shown that cancer patients with surgery (pooled HR = 0.46; 95%CI = 0.40–0.55; *P*<0.001) and other treatment strategies (pooled HR = 0.58; 95%CI = 0.50–0.68; *P*<0.001) were all associated with better OS. Moreover, cancer patients with higher AAPR after surgery (pooled HR = 0.464) had less risk of death than other treatment strategies (pooled HR = 0.58).

The corresponding cutoff values (range from 0.36 to 0.68) for AAPR were obtained from all included studies. The mean of the AAPR cutoff value was 0.5. Subsequently, all included studies were separated into two categories according to 0.5. There were seven studies in ≤ the 0.5 group, six studies in >0.5 group. The same results were obtained in two subgroups that higher AAPR predicted better OS (Group 1: pooled HR = 0.56, 95%CI = 0.48–0.65, *P*<0.001; Group 2: pooled HR = 0.49, 95%CI = 0.42–0.57, *P*<0.001).

## Meta-analysis on DFS

In three studies, the relation between DFS and AAPR was analyzed. A fixed effects model was applied due to non-significant heterogeneity ($I^2 = 0$, $P = 0.961$) (Fig 3). The results showed a correlation between higher AAPR and better DFS (pooled HR = 0.55; 95%CI = 0.47–0.66; *P*<0.001).

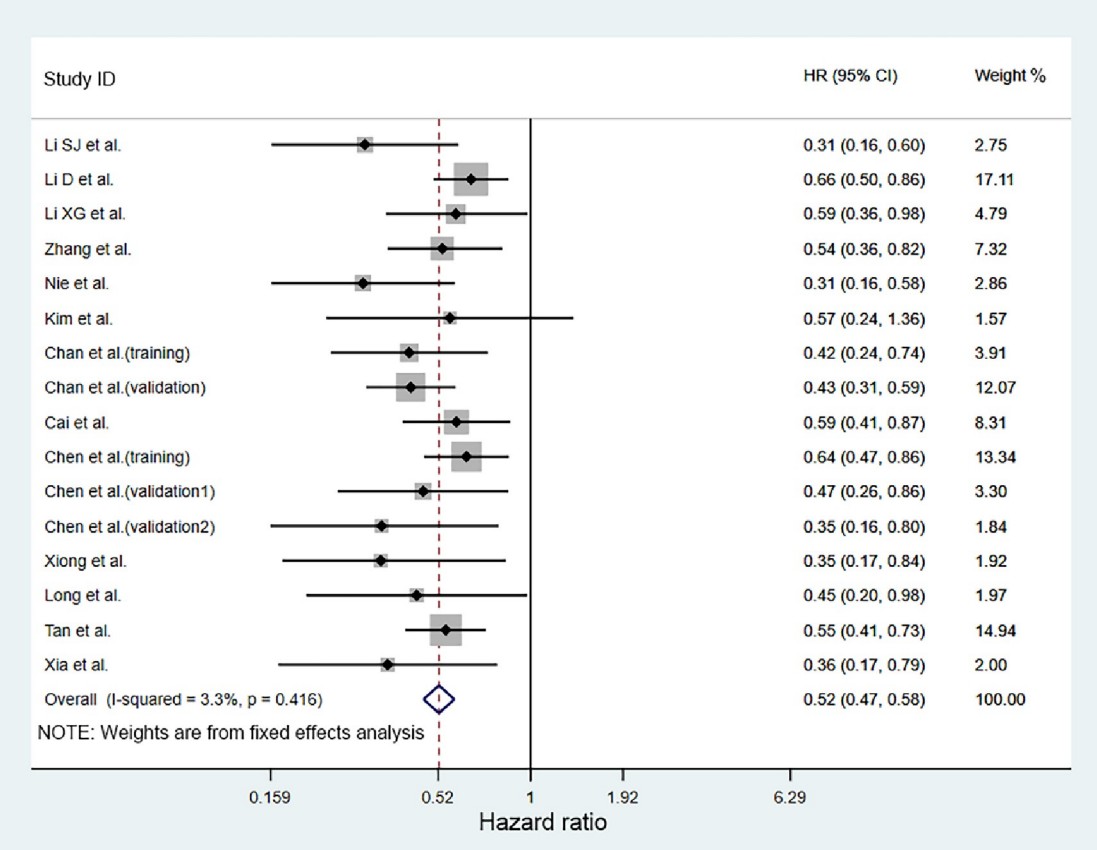

**Fig 2. Forest plot of hazard ratio for OS in cancer patients.**

**Table 2. Subgroup analysis for OS in the patients with cancer.**

| Stratified analysis | No. of studies | No. of patients | Effects model | HR (95% CI) | p | Heterogeneity | |
|---|---|---|---|---|---|---|---|
| | | | | | | $I^2$ (%) | p-value |
| **Cancer system** | | | | | | | |
| Respiratory | 4 | 1369 | Fixed | 0.58(0.48–0.70) | <0.001 | 32.6 | 0.22 |
| Digestive | 4 | 1669 | Fixed | 0.51(0.43–0.60) | <0.001 | 1.5 | 0.41 |
| Urinary | 2 | 1111 | Fixed | 0.52(0.40–0.68) | <0.001 | 0.0 | 0.33 |
| Others | 3 | 1055 | Fixed | 0.40(0.26–0.62) | <0.001 | 0.0 | 0.51 |
| **Treatment methods** | | | | | | | |
| Surgery | 7 | 3590 | Fixed | 0.46(0.40–0.55) | <0.001 | 0.0 | 0.72 |
| Others | 6 | 1614 | Fixed | 0.58(0.50–0.68) | <0.001 | 0.0 | 0.43 |
| **Cut off** | | | | | | | |
| ≤0.5 | 7 | 2214 | Fixed | 0.56(0.48–0.65) | <0.001 | 18.1 | 0.28 |
| >0.5 | 6 | 2990 | Fixed | 0.49(0.42–0.57) | <0.001 | 0.0 | 0.65 |

**Cancer system** Respiratory cancer: non-small-cell lung cancer (NSCLC), small-cell lung cancer (SCLC); Digestive cancer: hepatocellular carcinoma (HCC), cholangiocarcinoma (CCA); Urinary cancer: upper tract urothelial carcinoma (UTUC), renal cell carcinoma (RCC); Others: nasopharyngeal carcinoma (NPC), breast cancer (BC).

**Abbreviations:** HR, hazard ratio; CI, confidence interval.

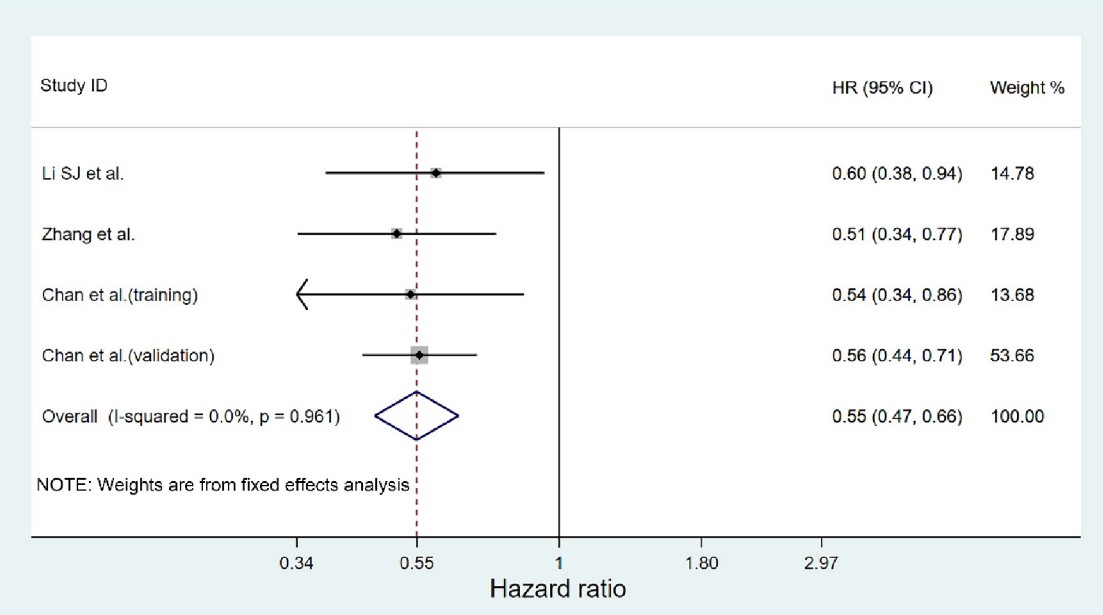

**Fig 3. Forest plot of hazard ratio for DFS in cancer patients.**

## Association between AAPR and clinical factors

A total of nine studies with 3,276 patients explored the relationship between AAPR and gender (Table 3). We adopted a fixed effects model as it is without significant heterogeneity ($I^2$ = 27.3%, $P$ = 0.20), S1 Fig. The results showed that AAPR was not significantly related to gender (pooled OR = 0.96; 95%CI = 0.86–1.07; $P$ = 0.43). Six articles with 2,867 patients covered the effect of AAPR on lymph node metastasis, S2 Fig. The fixed effects model was adopted ($I^2$ = 0.0%, $P$ = 0.51); as showed by the pooled results, the higher AAPR was associated with no lymph nodes metastasis (pooled OR = 0.89; 95%CI = 0.83–0.95; $P$ = 0.001). Five studies (2,150 patients) explored the correlation between AAPR and infiltration, S3 Fig. The higher AAPR was found to be associated with lighter infiltration (pooled OR = 0.79; 95%CI = 0.73–0.85; $P$<0.001). Only three studies (1,358 patients) investigated the association of AAPR with distant metastasis, S4 Fig. For patients with higher AAPR, there were more likely to be no distant metastasis (pooled OR = 0.92; 95%CI = 0.86–0.99; $P$ = 0.028), with no obvious heterogeneity ($I^2$ = 0.0%, $P$ = 0.60).

**Table 3. Results of the meta-analysis of clinicopathological characteristics.**

| Variables | No. of studies | No. of patients | Effects model | OR (95% CI) | p | Heterogeneity | |
|---|---|---|---|---|---|---|---|
| | | | | | | $I^2$(%) | p-value |
| **Sex (female vs. male)** | 9 | 3276 | Fixed | 0.96 (0.86–1.07) | 0.43 | 27.3 | 0.20 |
| **Lymph node metastasis (No vs. Yes)** | 6 | 2867 | Fixed | 0.89 (0.83–0.95) | 0.001 | 0.0 | 0.51 |
| **Infiltration (Tis-1-2 vs. T3-4)** | 5 | 2150 | Fixed | 0.79 (0.73–0.85) | <0.001 | 37.5 | 0.17 |
| **Distant metastasis (No vs. Yes)** | 3 | 1358 | Fixed | 0.92 (0.86–0.99) | 0.028 | 0.0 | 0.60 |

**Abbreviations:** OR, odds ratio; CI, confidence interval.

## Publication bias

Egger's test and Begg's funnel plot were applied to estimate the publication bias herein. Since, Begg's funnel plot was unsymmetrical (Fig 4A) with the *p*-value of Egger's test of 0.019, indicating that there was significant publication bias. Then, these problems were addressed by the "trim/fill method" (Fig 4B). Finally, the funnel plot was symmetric after being adjusted. Furthermore, the adjusted HR (pooled HR = 0.56; 95%CI = 0.51–0.62; *P* = 0.12) was consistent with that in the primary analysis (pooled HR = 0.52; 95%CI = 0.47–0.58; *P* = 0.42), which illustrated that the publication bias would not influence the reliability of the relationship between the low AAPR and poor OS.

## Analysis on sensitivity

Sensitivity analyses were further performed to investigate whether the pooled results would be affected by any single studies. At each step, one single study was omitted; the combined HRs for DFS/OS in cancer patients was not changed substantially, indicating that the meta-analysis was reliable and stable (Fig 5).

## Discussion

A novel risk factor (AAPR), calculated from ALB and ALP, was an inexpensive, non-invasive, and quickly acquired marker in clinical practice. In this meta-analysis, pooled data were collected from thirteen studies with 5,204 patients to evaluate AAPR's prognostic value in cancer patients comprehensively. These studies showed that the higher AAPR was associated with better OS and DFS (pooled HR = 0.52,0.55) in cancer patients. Subgroup analysis also indicated that AAPR over the cutoff value could forecast better OS in digestive cancer(*P*<0.001), respiratory cancer(*P*<0.001), urinary cancer(*P*<0.001), and other cancers (*P*<0.001). In subgroups of treatment methods, surgery (pooled HR = 0.46; 95%CI = 0.40–0.55; *P*<0.001) was the most favorable method that might result in better OS compared with other treatment strategies (pooled HR = 0.58; 95%CI = 0.50–0.68; *P*<0.001). Furthermore, the significant association between higher AAPR and clinicopathological features was also found in metastasis of lymph nodes (*P* = 0.001), infiltration (*P*<0.001), and distant metastasis (*P* = 0.028). Higher AAPR was not found to be associated with gender (*P* = 0.43).

Although the potential prognostic values of AAPR in cancer were not completely understood, there were several possible explanations. As for assessing the nutritional status, Albumin (ALB) was one of the most effective methods, which had a close association with immunity and inflammation. The increasing evidence showed that ALB could maintain DNA replication, promote cell proliferation, and modulate immune reactions [27]. Besides, ALB could exert antioxidant effects against carcinogens [28]. Therefore, the decreased ALB could reflect nutrient deficiency, which would lead to poor anti-cancer response and decline of immune function in cancer patients [27]. ALB was demonstrated in previous literature to be a valuable prognostic and predictive factor in renal carcinoma, prostate cancer, hepatocellular carcinoma (HCC), and other various cancers [29–31]. Alkaline phosphatase (ALP), a hydrolase converted in liver, kidneys, and bile duct, was commonly reported to link with bone metastasis, kidney and liver disease. Interestingly, the emerging evidence showed that ALP played a significant role in inflammation through adjustment of purinergic signaling, thus leading to the cessation of inflammatory signaling and causing an inhibitory immune response [32]. Mori et al. demonstrated that elevated ALP might reflect micrometastases, which could not be found on conventional imaging. This also partly explained why cancer patients with elevated ALP were more likely to have a poor prognosis. Meanwhile, cancer patients with elevated ALP would benefit more from intensive therapy than standard therapy [33]. Furthermore, ALP was also

A

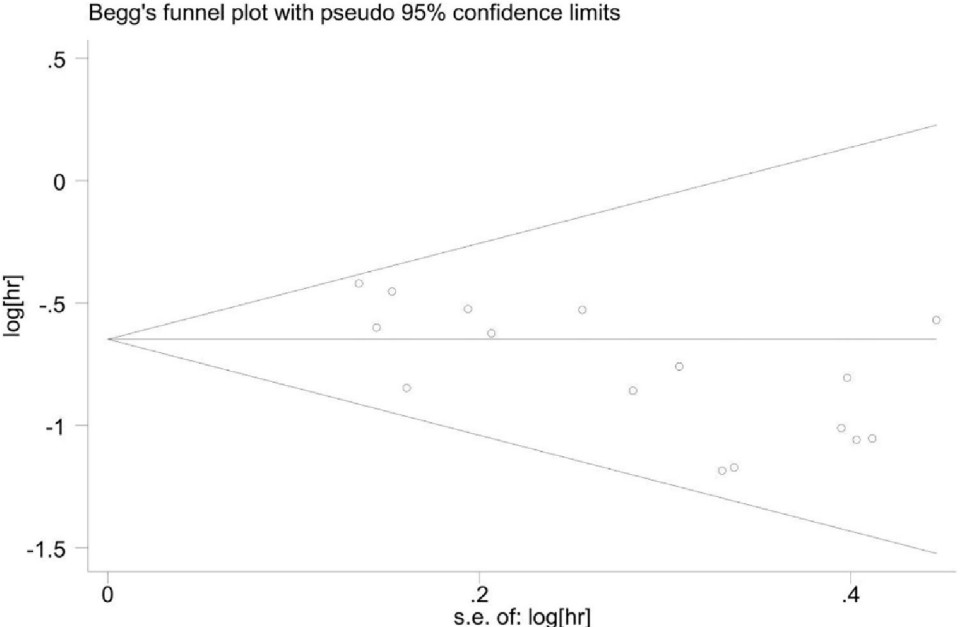

B

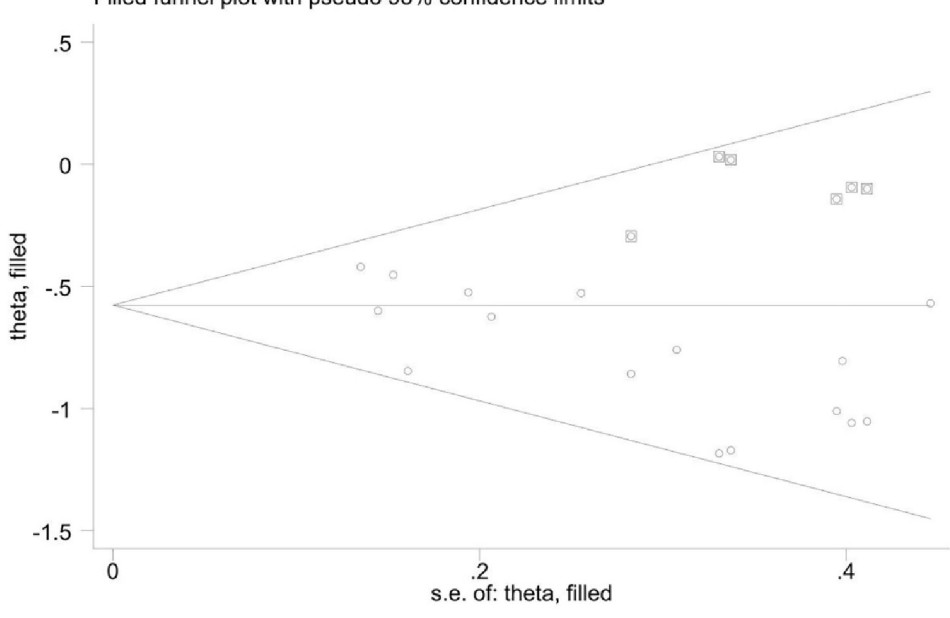

**Fig 4. Funnel plots for the evaluation of potential publication bias. A.** Funnel plots depicting the publication bias among the included studies on overall survival. **B.** The adjusted funnel plots depicting the publication bias among the included studies on overall survival.

A

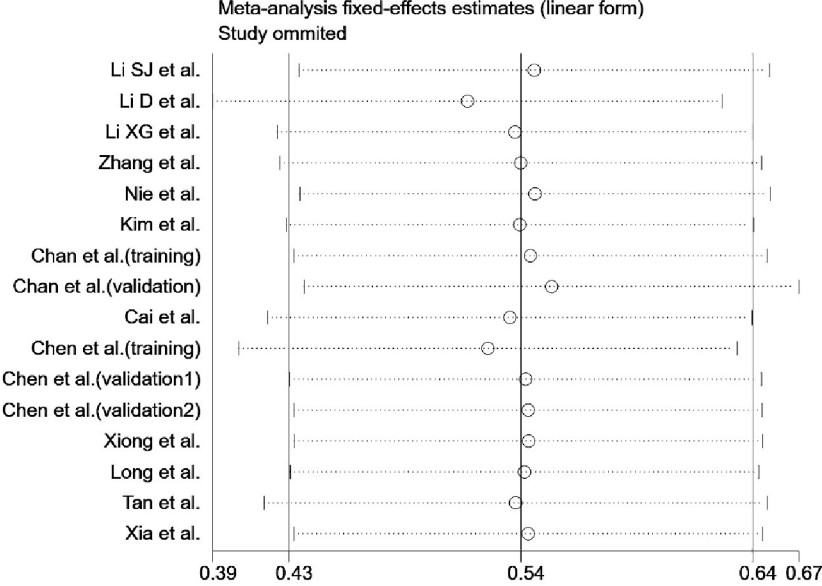

B

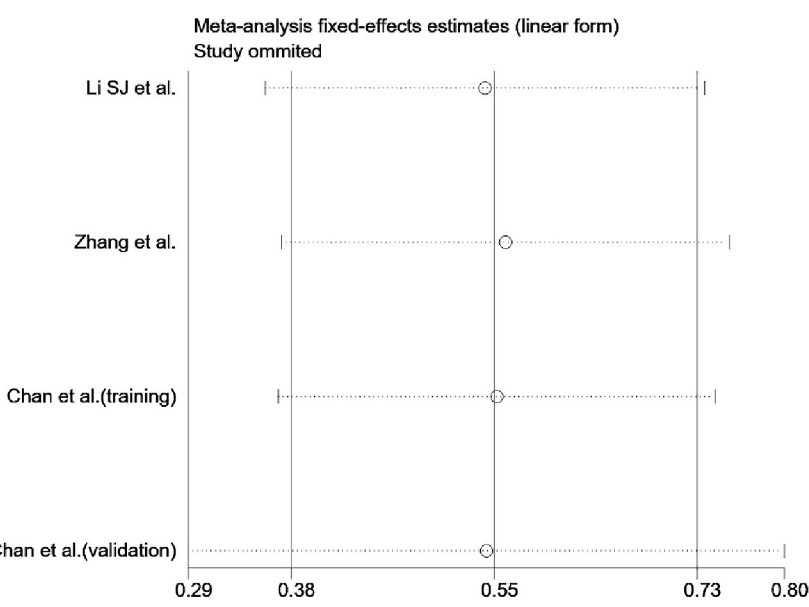

**Fig 5. Sensitivity analysis of our meta-analysis. A.** overall survival **B.** disease free survival.

expressed in cancer cells, which could regulate tumor growth [34]. Therefore, it could be concluded that elevated ALP indicated a worse survival in various cancers, including nasopharyngeal carcinoma, HCC, and RCC [35–37].

The concept of AAPR was firstly proposed by Anthony and colleagues, who proved that AAPR was a powerful and independent prognostic indicator with the highest $\chi^2$(by LR test)

and c-index compared with other biochemical parameters, such as albumin, alkaline phosphatase, alanine aminotransferase, and bilirubin [9]. As laboratory indexes, many conditions can interfere with serum albumin and alkaline phosphatase levels, including dehydration, fluid retention, illness, and pregnant, which may limit its credibility and application in the clinic. However, the combined laboratory index AAPR would be unaffected by many factors that affect the single albumin and alkaline phosphatase [38, 39]. Therefore, it was no doubt that the prognostic value of AAPR was more powerful than ALB or ALP alone. To our knowledge, it was the first meta-analysis focusing on AAPR's prognostic value. AAPR might help the clinicians distinguish the cancer patients with a high risk of the poor OS before the implementation of therapy. Patients with low AAPR might have hypoalbuminemia and increased ALP compared to patients with higher AAPR. That is to say, low AAPR can be used as a marker of decreased immunity, malnutrition, and increased treatment resistance [15]. The timely intervention of ALB in patients with low AAPR can increase AAPR, thereby correcting nutrition and improving treatment effectiveness. Meantime, patients with low AAPR might need more extra radiotherapy or chemotherapy, due to increased ALP might reflect micrometastases [33].

However, it should be acknowledged that there were several limitations to this meta-analysis. Firstly, the number of included studies and the sample size were both limited. Secondly, all these studies were carried out in two Asian countries, which may restrain their applicability. Thirdly, all these were retrospective studies reported in English, which would lead to potential biases. Fourthly, only three studies analyzed the relationship between AAPR and DFS, which should be further confirmed. Lastly, AAPR's cutoff values were different in these studies. Therefore, its prognostic value in cancer patients should be further investigated.

## Conclusion

In conclusion, this meta-analysis showed that the higher AAPR could be a significant and positive prognostic factor for better OS/DFS in various cancers. Nevertheless, further large-scale prospective and well-designed studies shall be conducted to confirm the prognostic value of AAPR.

## Supporting information

**S1 Table. PRISMA checklist.**
(DOC)

**S2 Table. Search strategies.**
(DOCX)

**S3 Table. Quality assessment of all included studies according to the Newcastle-Ottawa Scale.**
(DOCX)

**S1 Fig. Forest plot of the association between AAPR and sex (female versus male).**
(TIF)

**S2 Fig. Forest plot of the association between AAPR and lymph node metastasis (Yes versus No).**
(TIF)

**S3 Fig. Forest plot of the association between AAPR and in filtration (Tis-1-2 versus T3-4).**
(TIF)

**S4 Fig. Forest plot of the association between AAPR and distant metastasis (Yes versus No).**
(TIF)

## Author Contributions

**Conceptualization:** Hongmei Gu.

**Data curation:** Xiaoli Guo, Qijiu Zou, Jiaxin Yan, Xingxing Zhen.

**Formal analysis:** Xiaoli Guo, Qijiu Zou.

**Investigation:** Jiaxin Yan, Xingxing Zhen.

**Methodology:** Xiaoli Guo, Qijiu Zou.

**Project administration:** Hongmei Gu.

**Software:** Jiaxin Yan, Xingxing Zhen.

**Supervision:** Hongmei Gu.

**Validation:** Hongmei Gu.

**Visualization:** Hongmei Gu.

**Writing – original draft:** Qijiu Zou.

**Writing – review & editing:** Xiaoli Guo.

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
