## [Decision Letter · Decision Letter 0]

27 May 2020

PONE-D-20-05632

Prognostic effect of pretreatment albumin-to-alkaline phosphatase ratio in human cancers: a meta-analysis

PLOS ONE

Dear Dr. Gu,

Thank you for submitting your manuscript to PLOS ONE. After careful consideration, we feel that it has merit but does not fully meet PLOS ONE’s publication criteria as it currently stands. Therefore, we invite you to submit a revised version of the manuscript that addresses the points raised during the review process.

We look forward to receiving your revised manuscript.

Kind regards,

Jason Chia-Hsun Hsieh, M.D. Ph.D

Academic Editor

PLOS ONE

Additional Editor Comments:

The manuscript contains several problems, which bring the difficult decision for the manuscript.

1. The terminology and ways of interpretation seemed to be strange in the related professional fields. For example, the use of the random-effect model, an ideal OS, negative metastasis, lymph node metastasis, etc. require recheck.

2. There is no clinical use when there are no prospective trials conducted. Some interpretation for clinical use is too elaborate. For example, AAPR can help provide accurate and personalized therapy.

3. The unclear cutoff value of AAPR, some wrong HRs, and typos in the article limits the scientific values.

Please respond to the reviewers' comments in detail carefully.

Journal Requirements:

2. Please provide the complete search strategy for at least one database as a new supporting information file.

Reviewers' comments:

Reviewer's Responses to Questions

**Comments to the Author**

1. Is the manuscript technically sound, and do the data support the conclusions?

Reviewer #1: Yes

Reviewer #2: Partly

Reviewer #3: Yes

Reviewer #4: Yes

2. Has the statistical analysis been performed appropriately and rigorously? 

Reviewer #1: Yes

Reviewer #2: Yes

Reviewer #3: Yes

Reviewer #4: Yes

3. Have the authors made all data underlying the findings in their manuscript fully available?

Reviewer #1: Yes

Reviewer #2: Yes

Reviewer #3: Yes

Reviewer #4: Yes

4. Is the manuscript presented in an intelligible fashion and written in standard English?

Reviewer #1: Yes

Reviewer #2: No

Reviewer #3: Yes

Reviewer #4: No

5. Review Comments to the Author

Reviewer #1: In their manuscript, Guo and colleagues demonstrated that he albumin and alkaline 19 phosphatase ratio (AAPR) could be taken as a promising marker of prognosis. They analyzed the 5,204 cases of 8 types by using meta-analysis, and found that AAPR could result in better 27 DFS (HR=0.554, 95% CI: 0.465–0.659, p<0.001). Furthermore, the manuscript is well written and succinct.

Reviewer #2: The methodology of the meta-analysis is appropriate. However, there are several interpretations of the findings that could be problematic.

1. The authors stated several times that "high AAPR would ..." What is the threshold for "high" AAPR? Given that the cutoff values in the included literature varied, and the relative outcome measures (hazard ratio and odds ratio) were used, it may be better to say "higher AAPR would ..."

2. Line 97. "In order to determine the heterogeneity in these studies, a random-effect model was adopted."

Random-effect model is used when there is heterogeneity. Random-effect model is not used to determine the heterogeneity; heterogeneity is determined, for example, by I^2 and Q statistics as the authors stated.

3. Line 138. "...higher AAPR would lead to an ideal OS ..." What did "an ideal OS" refer to?

4. Association with other outcomes / clinical factors.

4a) Line 165 and 170. What is "negative metastasis"?

The authors stated that "high AAPR would result in ... without / no distant metastasis" (e.g. in Abstract and the Results section of the main text). This is too strong to say that high (enough) AAPR would lead to no distant metastasis.

4b) Line 160. Correlating AAPR and gender / infiltration.

Usually we correlate the measure (AAPR) with outcome (e.g., survival, presence of diseases) but not demographics.

Instead, the independence between AAPR and gender could be assessed by the heterogeneity between subgroups.

Similarly, "Infiltration (Tis-1–2 vs. T3-4)" is more the pretreatment prognosis rather than a post-treatment outcome?

Also, "A total of nine studies with 3,276 patients reported that there was an association between AAPR and gender (Table 3). They adopted a fixed effect model as it is without significant heterogeneity" Who adopted the fixed-effect model? Who "They" are?

5. Line 218. "There is no doubt that the prognostic value of AAPR is definitely more powerful than ALB or ALP alone."

What is the evidence for "AAPR is definitely more powerful" than ALB or ALP based on the stated findings?

Also, the authors quoted that lower ALB may result in poorer anti-cancer response (i.e., positive correlation). How about ALP? Whether it's positive or negative to anti-cancer response.

6. Line 220. "AAPR can help the clinicians to distinguish the cancer patients with a high risk of poor OS prior to the implementation of therapy, which could help to provide accurate and personalized therapy."

In which way AAPR can help provide accurate and personalized therapy? The authors may elaborate more.

7. Suggest to include the forest plots for other outcomes, putting as supplementary information would be sufficient.

8. Line 179-180. Why the HRs were negative?

9. Line 182. "Analysis on sensitivity" What kind of sensitivity were analysed?

10. Line 194. "Subgroup analysis also indicated that AAPR over the cutoff value could forecast better OS"

What was the "cutoff value"?

11. Figure 2.

The text of header can be put in one-line rather than in two lines.

Put more tick marks on the x-axis, not available even for the pooled HR. Not sure whether the x-axis is in linear or log scale.

12. Figure 3. The text is unreadable. Wonder whether the authors have checked the proofread PDF.

There is plenty of space for bigger font size.

Same issues as Figure 2 on the header and tick marks of x-axis.

13. Figure 5. The horizontal lines which show the estimates were almost invisible.

What does x-axis refer to? What do the two vertical lines at -0.76 and -0.54 refer to?

No need to assign background color for the plot.

Other comments.

1. Line 18-19. "... the albumin and alkaline phosphatase ratio (AAPR) for pretreatment is a prognostic factor". Perhaps it's better rephrased as "...the pretreatment albumin and alkaline phosphatase ratio (AAPR)..."

2. Line 25. "Results: In conclusions, this meta-analysis..." Not to use the term "In conclusions" in Results section, when this is not necessary.

3. Line 27. I think the "HRs" should be "pooled HRs", as to clarify that they were obtained from meta-analysis. Similar for "ORs" in the same paragraph.

4. Line 54. State the term of AAPR again when it's first used in the main text. Same for DFS, OS, HR, OR and CI, though they are well-known terms.

5. No need to capitalize the "carcinoma" and "meta-analysis" (e.g., top of page 10 and Table 1).

6. Cut-off or cutoff? Be consistent throughout the manuscript.

7. Line 100. What does AGR refer to?

8. Line 101-103. "Meta-analyses using variables including lymph node metastasis, infiltration and distant metastasis were conducted for further investigations."

lymph node metastasis and distant metastasis should be the outcomes rather than variables

9. Not to start sentences with digits, e.g., 448 potential articles were ...

10. Line 123. "Thirteen ones reported the relationship between AAPR and OS, and only three ones presented the association between AAPR and DFS."

Don't use the word "ones" when there is alternative available that will not confuse the readers.

For example, the authors could say "Thirteen studies / items / research / etc..."

11. Perhaps two decimal places are sufficient for non-statistically significant p-values and for HR / OR.

12. Line 190. "The results of these studies showed that the high AAPR would lead to better OS and DFS (HR=0.523, 0.554) in cancer patients, which means that cancer patients with low pretreatment AAPR would have lower OS/DFS due to the increased cancer recurrence/progression rate." The second statement (lower AAPR and lower OS/DFS) basically repeated the first statement (high AAPR and better OS / DFS).

13. Line 204. What was "ALB" referred to?

14. The authors are suggested to check the typos.

Reviewer #3: This study used meta-analysis focusing on the albumin and alkaline phosphatase ratio (AAPR) prognostic value. The results of this meta-analysis indicated that high AAPR could result in better DFS and OS. Moreover, high AAPR would result in lighter infiltration and negative metastasis of lymph nodes without distant metastasis. Despite the limitation of sample size and diversity, this study indicate that AAPR can help the clinicians to distinguish the cancer patients with a high risk of poor OS prior to the implementation of therapy, which could help to provide accurate and personalized therapy.

Here is a suggestion: Could the authors do the meta-analysis with albumin or alkaline phosphatase only in the same way, compare the results with AAPR and use them as kind of system control?

Reviewer #4: The authors present a meta-analysis of the prognostic effect of pretreatment albumin-to-alkaline phosphatase ratio in cancer patients. The meta-analysis included 13 studies, all conducted in Asia and found higher ratios associated with better overall survival (OS) and disease-free survival (DFS). The manuscript will be strengthened if the authors consider the following points.

1. The authors state that one of the inclusion criteria was that "there was an association of AAPR in serum with DFS and/or OS..." (lines 74-75). Do the authors mean to say this or do they mean to say something along the lines of "the association of AAPR in serum with DFS and/or OS was evaluated"? As stated in the manuscript, this would suggest that AAPR had to be significantly associated with survival in order to be included, which would be problematic if that was actually required.

2. Figure 1: authors say that 19 full-text articles were reviewed for eligibility and 7 were then excluded, which would be 12 articles eligible for inclusion instead of 13 - this should be clarified.

3. Table 2: a note should be included under the table to clarify for the reader which cancers fell into the different cancer systems - not all readers will be familiar with the different cancers.

4. Table 2: Why do the number of patients across the cancer systems not add to the total number of patients? All studies evaluated OS and since there is an "Others" category, all patients should be included.

5. Forest plots should be included in supplemental material for the results in Table 3 and the section beginning on line 159, since there is no information anywhere about which studies evaluated the clinical factors. The forest plots will allow the reader to have more information about the individual study results in addition to the presented combined results.

Minor points:

1. line 20: "of on cancer" should be "on cancer"

2. line 42: "were dead" - should this be "died"? I'm assuming that is what the authors meant.

3. line 47: "complexly disorder" should be "complex disorder"

4. line 56: I believe authors mean to say "predict" rather than "predicate"

5. line 56: authors should define OS (although they define it in the Abstract, it should be defined the first time it is used in the body of the manuscript. Similar for DFS (on line 75).

6. line 57: "evidences have showed" should be "evidence has shown"

7. line 59: "renal cell carcinom" should be "renal cell carcinoma"

8. line 69: "may be used" should be "were used" since this describes what the authors actually did.

9. Authors might consider including the details of the NOS for each study in a supplemental table so that readers know more about the quality of the studies.

10. line 100 - what is AGR?

11. line 108: STATA should be Stata (https://www.statalist.org/forums/help#spelling)

12. line 117: "literatures" should be "literature"

13. lines 123-124: replace "ones" with "studies"

14. line 142: "was showed" should be "was shown"

15. line 150: remove "Obviously"

16. Figure 3: the authors should increase the font size or do something about the clarity of the text in this figure. It is not as legible as the other figures.

17. line 161: rephrase the sentence starting with "They adopted..." as it is awkwardly phrased.

18. line 163: "coved" should be "covered"

19. line 165: "could lead to negative metastasis" should be rephrased - maybe something similar to "was associated with no metastasis"

20. line 167: remove "obviously"

21. line 168: change "ones" to "studies"

22. lines 179-180 - the reported HR and 95% CI are actually the betas and the corresponding CI (hazard ratios cannot be negative), so this should be clarified - the authors can transform the beta and CI to correspond to the HR, which might be easier for the reader to follow.

23. line 189: "maker" should be "marker"

24. line 196: the authors highlight surgery, but other the subgroup analysis for the other treatment strategies also showed a beneficial association of high AAPR on survival. There was just a sufficient number of studies to look specifically at surgery.

25. line 204: "evidences" should be "evidence" and authors should be consistent about abbreviations - here they use ALB, but they have also used Alb.

26. line 208: "literatures" should be"literature"

6. PLOS authors have the option to publish the peer review history of their article (what does this mean?). If published, this will include your full peer review and any attached files.

Reviewer #1: No

Reviewer #2: No

Reviewer #3: No

Reviewer #4: No

---

## [Author Response · Author response to Decision Letter 0]

8 Jul 2020

Response to Academic Editor: 

Question 1: The terminology and ways of interpretation seemed to be strange in the related professional fields. For example, the use of the random-effect model, an ideal OS, negative metastasis, lymph node metastasis, etc. require recheck.

Answer: We are very sorry for our mistakes. We have revised the manuscript as suggested.

(1) Line 100. "In order to determine the heterogeneity in these studies, a random-effect model was adopted (I2>50%, P<0.1); in other cases, the fixed-effect model should be used." was rewritten as "A random effects model was adopted when statistical heterogeneity was found (I2>50%, P<0.1); in other cases, the fixed effects model should be used."

(2) The "an ideal OS" referred to longer OS.

Line 141. " Furthermore, subgroup analyses of the included studies were conducted based on cancer system, treatment method and cutoff value (Table 2), which concluded that higher AAPR would lead to an ideal OS for patients with respiratory cancer (HR = 0.578; 95% CI: 0.477–0.702; P < 0.001), urinary cancer (HR = 0.523; 95% CI: 0.401–0.682; P < 0.001), digestive cancer (HR=0.518; 95% CI: 0.435–0.617; P<0.001), and other cancers (HR = 0.400; 95% CI: 0.259–0.617; P < 0.001)." was corrected as " Furthermore, subgroup analyses of the included studies were conducted based on cancer system, treatment method and cutoff value (Table 2), which concluded that higher AAPR would lead to a longer OS for patients with respiratory cancer (pooled HR=0.58; 95%CI:0.48–0.70; P<0.001), urinary cancer (pooled HR=0.52; 95%CI:0.40-0.68; P<0.001), digestive cancer (pooled HR=0.51; 95%CI=0.43-0.60; P<0.001), and other cancers (pooled HR=0.40; 95%CI=0.26-0.62; P<0.001)."

(3) The "negative metastasis" referred to no lymph nodes metastasis.

Line 25. "Moreover, high AAPR would result in lighter infiltration (OR: 0.79, 95% CI: 0.773-0.851, P<0.001) and negative metastasis of lymph nodes (OR: 0.89, 95% CI: 0.834-0.952, P<0.001) without distant metastasis (OR: 0.921, 95% CI: 0.856–0.996, P<0.001). " was revised as "Moreover, higher AAPR was statistically in associated with lighter infiltration (pooled OR=0.79; 95%CI=0.73-0.85; P<0.001), no lymph nodes metastasis (pooled OR=0.89; 95%CI=0.83-0.95; P=0.001), and no distant metastasis (pooled OR=0.92; 95%CI=0.86-0.99; P=0.028)."

Question 2: There is no clinical use when there are no prospective trials conducted. Some interpretation for clinical use is too elaborate. For example, AAPR can help provide accurate and personalized therapy.

Answer: We’re very sorry that we did not interpret clearly. We have revised the manuscript as suggested.

Line 244. "AAPR might help the clinicians distinguish the cancer patients with a high risk of the poor OS before the implementation of therapy, which could provide accurate and personalized treatment." was revised as "AAPR might help the clinicians distinguish the cancer patients with a high risk of the poor OS before the implementation of therapy."

Added content

(1) Line 30: "It might help physicians to select the most appropriate treatments by evaluating the current status of patients with cancer. Future multicenter prospective clinical trials were required to verify its applications."

(2) Line 245. "Patients with low AAPR might have hypoalbuminemia and increased ALP compared to patients with higher AAPR. That is to say, low AAPR can be used as a marker of decreased immunity, malnutrition, and increased treatment resistance.[15] The timely intervention of ALB in patients with low AAPR can increase AAPR, thereby correcting nutrition and improving treatment effectiveness. Meantime, patients with low AAPR might need more extra radiotherapy or chemotherapy, due to increased ALP might reflect micrometastases.[33] "

Question 3: The unclear cutoff value of AAPR, some wrong HRs, and typos in the article limits the scientific values.

Answer: We are very sorry for our mistakes. We have revised the manuscript as suggested and rechecked the entire manuscript carefully to avoid such mistakes.

Response to reviewers: 

Reviewer #1: In their manuscript, Guo and colleagues demonstrated that he albumin and alkaline 19 phosphatase ratio (AAPR) could be taken as a promising marker of prognosis. They analyzed the 5,204 cases of 8 types by using meta-analysis, and found that AAPR could result in better 27 DFS (HR=0.554, 95% CI: 0.465–0.659, p<0.001). Furthermore, the manuscript is well written and succinct.

Answer: Thank you very much for your comment.

Reviewer #2:

Question 1: The authors stated several times that "high AAPR would ..." What is the threshold for "high" AAPR? Given that the cutoff values in the included literature varied, and the relative outcome measures (hazard ratio and odds ratio) were used, it may be better to say "higher AAPR would ..."

Answer: Thank you very much for your suggestion. We have revised "high AAPR" to "higher AAPR" and checked the entire manuscript carefully to avoid such mistakes.

Question 2: Line 97. "In order to determine the heterogeneity in these studies, a random-effect model was adopted." Random-effect model is used when there is heterogeneity. Random-effect model is not used to determine the heterogeneity; heterogeneity is determined, for example, by I^2 and Q statistics as the authors stated.

Answer: We’re very sorry that we did not express clearly.

Line 100. "In order to determine the heterogeneity in these studies, a random-effect model was adopted (I2>50%, P<0.1); in other cases, the fixed-effect model should be used." was rewritten as "A random effects model was adopted when statistical heterogeneity was found (I2>50%, P<0.1); in other cases, the fixed effects model should be used."

Question 3: Line 138. "...higher AAPR would lead to an ideal OS ..." What did "an ideal OS" refer to?

Answer: We are very sorry for our inappropriate writing. The "an ideal OS" referred to longer OS.

Line 141. " Furthermore, subgroup analyses of the included studies were conducted based on cancer system, treatment method and cutoff value (Table 2), which concluded that higher AAPR would lead to an ideal OS for patients with respiratory cancer (HR = 0.578; 95% CI: 0.477–0.702; P < 0.001), urinary cancer (HR = 0.523; 95% CI: 0.401–0.682; P < 0.001), digestive cancer (HR=0.518; 95% CI: 0.435–0.617; P<0.001), and other cancers (HR = 0.400; 95% CI: 0.259–0.617; P < 0.001)." was corrected as " Furthermore, subgroup analyses of the included studies were conducted based on cancer system, treatment method and cutoff value (Table 2), which concluded that higher AAPR would lead to a longer OS for patients with respiratory cancer (pooled HR=0.58; 95%CI:0.48–0.70; P<0.001), urinary cancer (pooled HR=0.52; 95%CI:0.40-0.68; P<0.001), digestive cancer (pooled HR=0.51; 95%CI=0.43-0.60; P<0.001), and other cancers (pooled HR=0.40; 95%CI=0.26-0.62; P<0.001)."

Question 4: Association with other outcomes / clinical factors.

 Question 4a: Line 165 and 170. What is "negative metastasis"? The authors stated that "high AAPR would result in ... without / no distant metastasis" (e.g. in Abstract and the Results section of the main text). This is too strong to say that high (enough) AAPR would lead to no distant metastasis.

 Answer: We are very sorry for our inappropriate writing. The "negative metastasis" referred to no lymph nodes metastasis. We have revised the inappropriate writing in the Abstract and the Results section of the main text.

1. Line 25. "Moreover, high AAPR would result in lighter infiltration (OR: 0.79, 95% CI: 0.773-0.851, P<0.001) and negative metastasis of lymph nodes (OR: 0.89, 95% CI: 0.834-0.952, P<0.001) without distant metastasis (OR: 0.921, 95% CI: 0.856–0.996, P<0.001). " was revised as "Moreover, higher AAPR was statistically in associated with lighter infiltration (pooled OR=0.79; 95%CI=0.73-0.85; P<0.001), no lymph nodes metastasis (pooled OR=0.89; 95%CI=0.83-0.95; P=0.001), and no distant metastasis (pooled OR=0.92; 95%CI=0.86-0.99; P=0.028)."

2. Line 29. "High AAPR could result in better prognosis of cancer, and in cancer therapy, AAPR could be taken as a promising marker of prognosis." was corrected as " Higher AAPR was related to better prognosis of cancer, and in cancer therapy, AAPR could be taken as a promising marker of prognosis. "

3. Line 165. "The results showed that the high AAPR would result in a better DFS (HR = 0.554, 95% CI: 0.465–0.659, p < 0.001). " was revised as " The results showed a correlation between higher AAPR and better DFS (pooled HR=0.55; 95%CI=0.47-0.66; P<0.001)."

4. Line 173. "The fixed effect model was adopted (I2 =0.0%, P =0.513); as showed by the pooled results, high AAPR could lead to negative metastasis of lymph nodes (OR: 0.89, 95% CI: 0.834–0.952, P < 0.001). " was rewritten as "The fixed effects model was adopted (I2=0.0%, P=0.51); as showed by the pooled results, the higher AAPR was associated with no lymph nodes metastasis (pooled OR=0.89; 95%CI=0.83-0.95; P=0.001). "

5. Line178. " For patients with high AAPR, there would be no distant metastasis (OR: 0.921, 95% CI: 0.856–0.996, P<0.001), or obvious heterogeneity (I2 =0.0%, P =0.596) " was revised as " For patients with higher AAPR, there were more likely to be no distant metastasis (pooled OR=0.92; 95%CI= 0.86–0.99; P=0.028), with no obvious heterogeneity (I2=0.0%, P=0.60)."

 Question 4b: Line 160. Correlating AAPR and gender / infiltration. Usually we correlate the measure (AAPR) with outcome (e.g., survival, presence of diseases) but not demographics. Instead, the independence between AAPR and gender could be assessed by the heterogeneity between subgroups. Similarly, "Infiltration (Tis-1–2 vs. T3-4)" is more the pretreatment prognosis rather than a post-treatment outcome? Also, "A total of nine studies with 3,276 patients reported that there was an association between AAPR and gender (Table 3). They adopted a fixed effect model as it is without significant heterogeneity" Who adopted the fixed-effect model? Who "They" are?

 Answer: We are very sorry for our inappropriate writing. 

(1) We investigated the correlation between pretreatment AAPR and OS/DFS in Results. Next, we used pooled OR (odds ratio) and their 95%CI (confidence interval) to explore the relationship between AAPR and gender. Due to the lack of separate data corresponding to gender, we are sorry we can’t use the heterogeneity between subgroups to assess AAPR and gender. 

(2) The "Infiltration (Tis-1–2 vs. T3-4)" was more the pretreatment prognosis.

(3) Line 169. "A total of nine studies with 3,276 patients reported that there was an association between AAPR and gender (Table 3). They adopted a fixed effect model as it is without significant heterogeneity." was rewritten as " A total of nine studies with 3,276 patients explored the relationship between AAPR and gender (Table 3). We adopted a fixed effects model as it is without significant heterogeneity (I2=27.3%, P=0.202). "

Question 5a: Line 218. "There is no doubt that the prognostic value of AAPR is definitely more powerful than ALB or ALP alone." What is the evidence for "AAPR is definitely more powerful" than ALB or ALP based on the stated findings? 

Answer: Thank you very much for your question. We have added some content in Discussion.

1. Line 235. "The concept of AAPR was firstly proposed by Anthony and colleagues, who proved that AAPR was a powerful and independent prognostic indicator with the highestⅹ2(by LR test) and c-index compared with other biochemical parameters, such as albumin, alkaline phosphatase, alanine aminotransferase, and bilirubin. [9] " 

2. Line 238. "As laboratory indexes, many conditions can interfere with serum albumin and alkaline phosphatase levels, including dehydration, fluid retention, illness, and pregnant, which may limit its credibility and application in the clinic. However, the combined laboratory index AAPR would be unaffected by many factors that affect the single albumin and alkaline phosphatase. [38-39] " 

Question 5b: Also, the authors quoted that lower ALB may result in poorer anti-cancer response (i.e., positive correlation). How about ALP? Whether it's positive or negative to anti-cancer response.

Answer: Thank you very much for your question. 

(1) Many studies have identified that elevated ALP level was not only correlated with kidney, liver, and bone diseases but also indicated a worse survival in lung and colon cancer patients. [1-3] 

References

[1] Hung HY, Chen JS, Chien Y, Tang R, Hsieh PS, Wen S, et al. Preoperative alkaline phosphatase elevation was associated with poor survival in colorectal cancer patients. Int J Colorectal Dis. 2017;32(12):1775-8. Epub 2017/10/17. doi: 10.1007/s00384-017-2907-4. PMID: 29030683.

[2] Schoppet M, Shanahan CM. Role for alkaline phosphatase as an inducer of vascular calcification in renal failure? Kidney Int. 2008;73(9):989-91. Epub 2008/04/17. doi: 10.1038/ki.2008.104. PMID: 18414436.

[3] Li SJ, Lv WY, Du H, Li YJ, Zhang WB, Che GW, et al. Albumin-to-alkaline phosphatase ratio as a novel prognostic indicator for patients undergoing minimally invasive lung cancer surgery: Propensity score matching analysis using a prospective database. Int J Surg. 2019;69:32-42. Epub 2019/07/19. doi: 10.1016/j.ijsu.2019.07.008. PMID: 31319230.

(2) And we added some content in Discussion.

Line 227. "Mori et al. demonstrated that elevated ALP might reflect micrometastases, which could not be found on conventional imaging. This also partly explained why cancer patients with elevated ALP were more likely to have a poor prognosis. Meanwhile, cancer patients with elevated ALP would benefit more from intensive therapy than standard therapy.[33]"

Question 6: Line 220. "AAPR can help the clinicians to distinguish the cancer patients with a high risk of poor OS prior to the implementation of therapy, which could help to provide accurate and personalized therapy."

In which way AAPR can help provide accurate and personalized therapy? The authors may elaborate more.

Answer: Thank you very much for your advice. We have added some content in Discussion.

Line 245. "Patients with low AAPR might have hypoalbuminemia and increased ALP compared to patients with higher AAPR. That is to say, low AAPR can be used as a marker of decreased immunity, malnutrition, and increased treatment resistance.[15] The timely intervention of ALB in patients with low AAPR can increase AAPR, thereby correcting nutrition and improving treatment effectiveness. Meantime, patients with low AAPR might need more extra radiotherapy or chemotherapy, due to increased ALP might reflect micrometastases.[33] "

Question 7: Suggest to include the forest plots for other outcomes, putting as supplementary information would be sufficient.

Answer: Thank you very much for your suggestion. But only two articles of all the included studies had the prognostic date of PFS (progression-free survival), CSS (cancer-specific survival), and RFS (recurrent-free survival). Meanwhile, the other outcomes in the include studies were too less. Therefore, the forest plots based on these outcomes would be unreliable. 

Question 8: Line 179-180. Why the HRs were negative?

Answer: We are very sorry for our mistake. 

Log [HR1]=-0.576 ; Log [HR2]=-0.648

HR1=0.562 ; HR2=0.523

Line189. "Furthermore, the adjusted HR (HR: -0.576, 95% CI: -0.678~-0.474, P=0.124) was consistent with that in the primary analysis (HR: -0.648, 95% CI: -0.757~-0.538, P=0.416)." was corrected as "Furthermore, the adjusted HR (HR= 0.56; 95%CI=0.51-0.62; P=0.12) was consistent with that in the primary analysis (HR=0.52; 95%CI:0.47-0.584; P=0.42), which illustrated that the publication bias would not influence the reliability of the relationship between low AAPR and poor OS."

Question 9: Line 182. "Analysis on sensitivity" What kind of sensitivity were analyzed?

Answer: Thank you very much for your question. Sensitivity analyses were further performed to investigate whether the pooled results would be affected by any single studies. It was carried out by sequentially omitting individual studies at each step. If the results did not substantially alter when one study was excluded, this meant that the pooled results were stable.

Line 195. "Sensitivity analyses were further performed to investigate whether the pooled results would be affected by any single studies."

Question 10: Line 194. "Subgroup analysis also indicated that AAPR over the cutoff value could forecast better OS" What was the "cutoff value"?

Answer: Thank you very much for your question. We have revised our manuscript. The cutoff value here referred to the mean of the AAPR cutoff value (0.5).

Line156. " The corresponding cutoff values (range from 0.36 to 0.68) for AAPR were obtained from all included studies. The mean of the AAPR cutoff value was 0.5. Subsequently, all included studies were separated into two categories according to 0.5. There were seven studies in ≤ the 0.5 group, six studies in >0.5 group." 

Question 11: Figure 2. The text of header can be put in one-line rather than in two lines. Put more tick marks on the x-axis, not available even for the pooled HR. Not sure whether the x-axis is in linear or log scale.

Answer: Thank you very much for your comment. We have revised Figure2. as suggested. The x-axis was log scale.

Figure 2. Forest plot of hazard ratio for OS in cancer patients.

Question 12: Figure 3. The text is unreadable. Wonder whether the authors have checked the proofread PDF. There is plenty of space for bigger font size. Same issues as Figure 2 on the header and tick marks of x-axis.

Answer: We are very sorry for our mistake. We have revised Figure3. as suggested.

Figure 3. Forest plot of hazard ratio for DFS in cancer patients.

Question 13: Figure 5. The horizontal lines which show the estimates were almost invisible.

What does x-axis refer to? What do the two vertical lines at -0.76 and -0.54 refer to? No need to assign background color for the plot.

Answer: We are very sorry for our mistakes. The x-axis referred to ln(HR) in the original Figure 5, leading to the x-axis was negative. We have revised Figure 5. Now the x-axis refers to HR, and the two vertical lines indicate to 95%CI (confidence interval).

Figure 5. Sensitivity analysis test for overall survival.

Other comments.

Question 1: Line 18-19. "... the albumin and alkaline phosphatase ratio (AAPR) for pretreatment is a prognostic factor". Perhaps it's better rephrased as "...the pretreatment albumin and alkaline phosphatase ratio (AAPR)..."

Answer: Thank you very much for your comment. We have revised the manuscript as suggested.

Question 2：Line 25. "Results: In conclusions, this meta-analysis..." Not to use the term "In conclusions" in Results section, when this is not necessary.

Answer: Thank you very much for your comment. We have revised the manuscript as suggested.

Question 3：Line 27. I think the "HRs" should be "pooled HRs", as to clarify that they were obtained from meta-analysis. Similar for "ORs" in the same paragraph.

Answer: Thank you very much for your comment. We have revised the manuscript as suggested.

Question 4：Line 54. State the term of AAPR again when it's first used in the main text. Same for DFS, OS, HR, OR and CI, though they are well-known terms.

Answer: Thank you very much for your comment. We have revised the manuscript as suggested.

Question 5：No need to capitalize the "carcinoma" and "meta-analysis" (e.g., top of page 10 and Table 1).

Answer: Thank you very much for your comment. We have revised the manuscript as suggested.

Question 6：Cut-off or cutoff? Be consistent throughout the manuscript.

Answer: Thank you very much for your comment. We have revised the manuscript as suggested.

Question 7：Line 100. What does AGR refer to?

Answer: We are very sorry for our incorrect writing. 

Line 102. " HR<1 (high AGR used as reference) showed a lower risk of worse outcomes for high AAPR; meanwhile, if P<0.05 and 95% CI <1, it would be deemed as statistically significant." was corrected as " HR<1 (higher AAPR used as reference) showed a lower risk of worse outcomes for higher AAPR; meanwhile, if P<0.05 and 95%CI <1, it would be deemed as statistically significant."

Question 8：Line 101-103. "Meta-analyses using variables including lymph node metastasis, infiltration and distant metastasis were conducted for further investigations." lymph node metastasis and distant metastasis should be the outcomes rather than variables

Answer: We are very sorry for our mistakes.

Line 104. "Meta-analyses using variables including lymph node metastasis, infiltration and distant metastasis were conducted for further investigations. " was corrected as "Pooled ORs (odds ratios) and related 95%CI were used to estimate the relationship between AAPR and clinical features, including lymph node metastasis, infiltration, and distant metastasis."

Question 9：Not to start sentences with digits, e.g., 448 potential articles were ...

Answer: Thank you very much for your comment. We have revised the manuscript as suggested.

Line 116. "448 potential articles were collected through retrieving Web of Science, PubMed and Embase. 105 articles were removed due to duplication. " was rephrased as "A total of 448 potential articles were collected through retrieving Web of Science, PubMed, and Embase. Among these studies, 105 articles were removed due to duplication. "

Question 10：Line 123. "Thirteen ones reported the relationship between AAPR and OS, and only three ones presented the association between AAPR and DFS."

Don't use the word "ones" when there is alternative available that will not confuse the readers.

For example, the authors could say "Thirteen studies / items / research / etc..."

Answer: We are very sorry for our mistakes. We have revised the manuscript as suggested.

Line 127. "Thirteen ones reported the relationship between AAPR and OS, and only three ones presented the association between AAPR and DFS." was revised as " Thirteen studies reported the relationship between AAPR and OS, while only three studies presented the association between AAPR and DFS."

Question 11：Perhaps two decimal places are sufficient for non-statistically significant p-values and for HR / OR

Answer: Thank you very much for your comment. We have revised the manuscript as suggested.

Question 12：Line 190. "The results of these studies showed that the high AAPR would lead to better OS and DFS (HR=0.523, 0.554) in cancer patients, which means that cancer patients with low pretreatment AAPR would have lower OS/DFS due to the increased cancer recurrence/progression rate." The second statement (lower AAPR and lower OS/DFS) basically repeated the first statement (high AAPR and better OS / DFS).

Answer: Thank you very much for your comment. We have revised the manuscript as suggested.

Line 204. "The results of these studies showed that the high AAPR would lead to better OS and DFS (HR=0.523, 0.554) in cancer patients, which means that cancer patients with low pretreatment AAPR would have lower OS/DFS due to the increased cancer recurrence/progression rate." was rewritten as "These studies showed that the higher AAPR was associated with better OS and DFS (pooled HR=0.52, 0.55) in cancer patients."

Question 13：Line 204. What was "ALB" referred to?

Answer: We are sorry for our mistakes. The "ALB" referred to albumin.

Line 215. "As for the assessment of the nutritional status, Albumin(A) was one of the most effective methods, which had a close association with immunity and inflammation." was corrected as " As for assessing the nutritional status, Albumin (ALB) was one of the most effective methods, which had a close association with immunity and inflammation."

Question 14：The authors are suggested to check the typos.

Answer: We are sorry for our mistakes and thank you for your reminder. We have reviewed the entire manuscript carefully to avoid such mistakes.

Reviewer #3:

Question 1: This study used meta-analysis focusing on the albumin and alkaline phosphatase ratio (AAPR) prognostic value. The results of this meta-analysis indicated that high AAPR could result in better DFS and OS. Moreover, high AAPR would result in lighter infiltration and negative metastasis of lymph nodes without distant metastasis. Despite the limitation of sample size and diversity, this study indicate that AAPR can help the clinicians to distinguish the cancer patients with a high risk of poor OS prior to the implementation of therapy, which could help to provide accurate and personalized therapy.

Here is a suggestion: Could the authors do the meta-analysis with albumin or alkaline phosphatase only in the same way, compare the results with AAPR and use them as kind of system control?

Answer: Thank you very much for your suggestion. 

(1) As laboratory indexes, many conditions can interfere with serum albumin and alkaline phosphatase levels, including dehydration, fluid retention, illness, and pregnant, which may limit its credibility and application in the clinic. However, the combined laboratory index AAPR was more reliable, owing to it could minimize the potential bias. [1,2]

(2) The concept of AAPR was firstly proposed by Anthony and colleagues, who proved that AAPR was a powerful and independent prognostic indicator with the highestⅹ2(by LR test) and c-index compared with albumin and alkaline phosphatase.[3]

(3) Some studies have reported that AAPR was an independent prognostic factor associated with OS/DFS than albumin and alkaline phosphatase in multivariate survival analysis. [4,5]

In conclusion, as a prognostic indicator, AAPR has many advantages in cancer patients compared with the other two indices.

Reference:

[1] Bizzo SM, Meira DD, Lima JM, Mororo Jda S, Moreira FC, Casali-da-Rocha JC, et al. Serum albumin and vascular endothelial growth factor in epithelial ovarian cancer: looking at adnexal tumor drainage. Arch Gynecol Obstet. 2011;283(4):855-9. Epub 2010/05/12. doi: 10.1007/s00404-010-1491-4. PMID: 20458489.

[2] Boronkai A, Than NG, Magenheim R, Bellyei S, Szigeti A, Deres P, et al. Extremely high maternal alkaline phosphatase serum concentration with syncytiotrophoblastic origin. J Clin Pathol. 2005;58(1):72-6. Epub 2004/12/30. doi: 10.1136/jcp.2003.015362. PMID: 15623487.

[3] Chan AW, Chan SL, Mo FK, Wong GL, Wong VW, Cheung YS, et al. Albumin-to-alkaline phosphatase ratio: a novel prognostic index for hepatocellular carcinoma. Dis Markers. 2015;2015:564057. Epub 2015/03/05. doi: 10.1155/2015/564057. PMID: 25737613.

[4] Zhang L, Zhang H, Yue D, Wei W, Chen Y, Zhao X, et al. The prognostic value of the preoperative albumin to alkaline phosphatase ratio in patients with non-small cell lung cancer after surgery. Thorac Cancer. 2019;10(7):1581-9. Epub 2019/06/05. doi: 10.1111/1759-7714.13107. PMID: 31161711.

[5] Li SJ, Lv WY, Du H, Li YJ, Zhang WB, Che GW, et al. Albumin-to-alkaline phosphatase ratio as a novel prognostic indicator for patients undergoing minimally invasive lung cancer surgery: Propensity score matching analysis using a prospective database. Int J Surg. 2019;69:32-42. Epub 2019/07/19. doi: 10.1016/j.ijsu.2019.07.008. PMID: 31319230

Reviewer #4:

Question 1: The authors state that one of the inclusion criteria was that "there was an association of AAPR in serum with DFS and/or OS..." (lines 74-75). Do the authors mean to say this or do they mean to say something along the lines of "the association of AAPR in serum with DFS and/or OS was evaluated"? As stated in the manuscript, this would suggest that AAPR had to be significantly associated with survival in order to be included, which would be problematic if that was actually required.

Answer: We are very sorry for our inappropriate writing. We mean that the included articles should investigate the relationship between AAPR and cancer prognosis, whether they were significantly relevant or not. 

Line 75. "In these studies, the inclusion criteria were adopted as follows: (1) There was an association of AAPR in serum with DFS and/or OS of primary cancer; (2) HRs and 95% CIs for the prognosis were provided; (3) The cut-off value of AAPR had been reported; (4) The text was prepared in English." was rephrased as "In these studies, the inclusion criteria were adopted as follows: (1) articles investigated the association with AAPR and cancer prognosis; (2) HRs(hazard ratios) and 95% CIs(confidence intervals) for the prognosis were provided; (3) the cutoff value of AAPR had been reported; and (4) the text was prepared in English."

Question 2: Figure 1: authors say that 19 full-text articles were reviewed for eligibility and 7 were then excluded, which would be 12 articles eligible for inclusion instead of 13 - this should be clarified.

Answer: We are very sorry for our mistakes. A flow diagram regarding literature retrieval was expressed in the Figure 1. A total of 448 potential articles were collected through retrieving Web of Science, PubMed, and Embase. 105 articles were removed due to duplication. After the titles and abstracts had been screened by two investigators, 324 studies were excluded. Subsequently, 19 full-text articles were reviewed for eligibility. Finally, this meta-analysis involved 13 studies in total (5,204 patients).

Figure 1. A flow diagram for the literature assessment process

Question 3: Table 2: a note should be included under the table to clarify for the reader which cancers fell into the different cancer systems - not all readers will be familiar with the different cancers.

Answer: Thank you very much for your comment. We have added a note under Table 2.

Line 152. "Cancer system Respiratory cancer: non-small-cell lung cancer (NSCLC), small-cell lung cancer (SCLC); Digestive cancer: hepatocellular carcinoma (HCC), cholangiocarcinoma (CCA); Urinary cancer: upper tract urothelial carcinoma (UTUC), renal cell carcinoma (RCC); Others: nasopharyngeal carcinoma (NPC), breast cancer (BC)."

Question 4: Table 2: Why do the number of patients across the cancer systems not add to the total number of patients? All studies evaluated OS and since there is an "Others" category, all patients should be included.

Answer: We are very sorry for our mistakes. We have corrected Table 2.

Table 2. Subgroup analysis for OS in the patients with cancer

Stratified analysis No. of No. of Effects HR (95% CI) p Heterogeneity

 studies patients model I² (%) p-value

Cancer system 

 Respiratory 4 1369 Fixed 0.58(0.48-0.70) <0.001 32.6 0.217

 Digestive 4 1669 Fixed 0.51(0.43-0.60) <0.001 1.5 0.413

 Urinary 2 1111 Fixed 0.52(0.40-0.68) <0.001 0.0 0.329

 Others 3 1055 Fixed 0.40(0.26-0.62) <0.001 0.0 0.512

Treatment methods 

 Surgery 7 3590 Fixed 0.46(0.40-0.55) <0.001 0.0 0.719

 Others 6 1614 Fixed 0.58(0.50-0.68) <0.001 0.0 0.431

Cut off 

 ≤0.5 7 2214 Fixed 0.56(0.48-0.65) <0.001 18.1 0.281

>0.5 6 2990 Fixed 0.49(0.42-0.57) <0.001 0.0 0.651

Cancer system Respiratory cancer: non-small-cell lung cancer (NSCLC), small-cell lung cancer (SCLC); Digestive cancer: hepatocellular carcinoma (HCC), cholangiocarcinoma (CCA); Urinary cancer: upper tract urothelial carcinoma (UTUC), renal cell carcinoma (RCC); Others: nasopharyngeal carcinoma (NPC), breast cancer (BC).

Abbreviations: HR, hazard ratio; CI, confidence interval.

Question 5：Forest plots should be included in supplemental material for the results in Table 3 and the section beginning on line 159, since there is no information anywhere about which studies evaluated the clinical factors. The forest plots will allow the reader to have more information about the individual study results in addition to the presented combined results.

Answer: Thank you very much for your comment. We have added the forest plots for Table 3 in supplemental materials.

S1 Fig. Forest plot of the association between AAPR and sex (female versus male).

S2 Fig. Forest plot of the association between AAPR and lymph node metastasis (No versus Yes).

S3 Fig. Forest plot of the association between AAPR and infiltration (Tis-1-2 versus T3-4).

S4 Fig. Forest plot of the association between AAPR and distant metastasis (No versus Yes).

Minor points:

Question 1：line 20: "of on cancer" should be "on cancer"

Answer: Thank you very much for your comment. We have revised the manuscript as suggested.

Question 2：line 42: "were dead" - should this be "died"? I'm assuming that is what the authors meant.

Answer: We’re very sorry that we did not express clearly.

Line 39："In the United States, 1,762,450 new cancer patients were estimated in 2019, 606,880 of whom were dead.[1]" was revised as " It was estimated that there were 1,762,450 new cancer cases and 606,880 deaths in the United States in 2019.[1]" 

Question 3：line 47: "complexly disorder" should be "complex disorder"

Answer: We are very sorry for our mistakes. We have revised the manuscript as suggested.

Question 4：line 56: I believe authors mean to say "predict" rather than "predicate"

Answer: We are very sorry for our mistakes. We have revised the manuscript as suggested.

Question 5：line 56: authors should define OS (although they define it in the Abstract, it should be defined the first time it is used in the body of the manuscript. Similar for DFS (on line 75).

Answer: Thank you very much for your comment. We have revised the manuscript as suggested.

Question 6：line 57: "evidences have showed" should be "evidence has shown"

Answer: We are very sorry for our mistakes. We have revised the manuscript as suggested.

Question 7：line 59: "renal cell carcinom" should be "renal cell carcinoma"

Answer: We are very sorry for our mistakes. We have revised the manuscript as suggested

Question 8：line 69: "may be used" should be "were used" since this describes what the authors actually did.

Answer: We are very sorry for our mistakes. We have revised the manuscript as suggested.

Question 9：Authors might consider including the details of the NOS for each study in a supplemental table so that readers know more about the quality of the studies.

Answer: Thank you very much for your comment. We have added S3 table for the details of the NOS for all included studies in Supporting information.

Question 10：line 100 - what is AGR?

Answer: We are very sorry for our incorrect writing. 

Line 102. " HR<1 (high AGR used as reference) showed a lower risk of worse outcomes for high AAPR; meanwhile, if P<0.05 and 95% CI <1, it would be deemed as statistically significant." was corrected as " HR<1 (higher AAPR used as reference) showed a lower risk of worse outcomes for higher AAPR; meanwhile, if P<0.05 and 95% CI <1, it would be deemed as statistically significant."

Question 11：line 108: STATA should be Stata (https://www.statalist.org/forums/help#spelling)

Answer: Thank you very much for your comment. We have revised the manuscript as suggested.

Question 12：line 117: "literatures" should be "literature"

Answer: We are very sorry for our mistakes. We have revised the manuscript as suggested.

Question 13：lines 123-124: replace "ones" with "studies"

Answer: We are very sorry for our mistakes. We have revised the manuscript as suggested.

Question 14：line 142: "was showed" should be "was shown"

Answer: We are very sorry for our mistakes. We have revised the manuscript as suggested.

Question 15：line 150: remove "Obviously"

Answer: Thank you very much for your comment. We have revised the manuscript as suggested.

Question 16：Figure 3: the authors should increase the font size or do something about the clarity of the text in this figure. It is not as legible as the other figures.

Answer: Thank you very much for your comment. We have revised Figure 3 as suggested.

Figure 3. Forest plot of hazard ratio for DFS in cancer patients.

Question 17：line 161: rephrase the sentence starting with "They adopted..." as it is awkwardly phrased.

Answer: We are very sorry for our mistakes. We have revised the manuscript as suggested.

Line 170. "They adopted a fixed effect model as it is without significant heterogeneity (I2=27.3%, P =0.202)." was rephrased as "We adopted a fixed effects model as it is without significant heterogeneity (I2=27.3%, P=0.20)"

Question 18：line 163: "coved" should be "covered"

Answer: We are very sorry for our mistakes. We have revised the manuscript as suggested.

Question 19：line 165: "could lead to negative metastasis" should be rephrased - maybe something similar to "was associated with no metastasis"

Answer: We are very sorry for our mistakes. We have revised the manuscript as suggested.

Line 173. "The fixed effect model was adopted (I2=0.0%, P =0.513); as showed by the pooled results, high AAPR could lead to negative metastasis of lymph nodes (OR:0.89, 95%CI:0.834–0.952, P<0.001)." was revised as "The fixed effects model was adopted (I2=0.0%, P=0.51); as showed by the pooled results, the higher AAPR was associated with no lymph nodes metastasis (pooled OR=0.89; 95%CI:0.83-0.95; P<0.001).

Question 20：line 167: remove "obviously"

Answer: We are very sorry for our mistakes. We have revised the manuscript as suggested.

Question 21：line 168: change "ones" to "studies"

Answer: We are very sorry for our mistakes. We have revised the manuscript as suggested.

Question 22：lines 179-180 - the reported HR and 95% CI are actually the betas and the corresponding CI (hazard ratios cannot be negative), so this should be clarified - the authors can transform the beta and CI to correspond to the HR, which might be easier for the reader to follow.

Answer: We are very sorry for our mistakes. We have revised the manuscript as suggested.

Line 189. " Furthermore, the adjusted HR (HR: -0.576, 95% CI: -0.678~-0.474, P=0.124) was consistent with that in the primary analysis (HR: -0.648, 95% CI: -0.757~-0.538, P=0.416)." was corrected as "Furthermore, the adjusted HR (pooled HR=0.56; 95%CI=0.51-0.62; P=0.12) was consistent with that in the primary analysis (pooled HR=0.52; 95%CI=0.47-0.58; P=0.42), which illustrated that the publication bias would not influence the reliability of the relationship between the low AAPR and poor OS."

Question 23：line 189: "maker" should be "marker"

Answer: We are very sorry for our mistakes. We have revised the manuscript as suggested.

Question 24：line 196: the authors highlight surgery, but other the subgroup analysis for the other treatment strategies also showed a beneficial association of high AAPR on survival. There was just a sufficient number of studies to look specifically at surgery.

Answer: Thank you very much for your comment. We have revised the manuscript as suggested.

Line 146. "For the subgroup involving the treatment method, it was shown that higher AAPR was a significant favorable factor in patients with cancer after surgery (pooled HR=0.46; 95%CI=0.40-0.55; P<0.001)." was revised as "For the subgroup involving the treatment method, it was shown that cancer patients with surgery (pooled HR=0.464; 95%CI=0.40-0.55; P<0.001) and other treatment strategies (pooled HR=0.58; 95%CI=0.50-0.68; P<0.001) were all associated with better OS. Moreover, cancer patients with higher AAPR after surgery (pooled HR=0.464) had less risk of death than other treatment strategies (pooled HR=0.58)."

Line 207. "In subgroups of treatment methods, surgery (HR: 0.464, 95% CI: 0.396 - 0.545，P<0.001) was the most positive method that could result in better OS." was corrected as "In subgroups of treatment methods, surgery (pooled HR=0.46; 95%CI=0.40-0.55; P<0.001) was the most favorable method that might result in better OS compared with other treatment strategies (pooled HR=0.58; 95%CI=0.50-0.68; P<0.001)."

Question 25：line 204: "evidences" should be "evidence" and authors should be consistent about abbreviations - here they use ALB, but they have also used Alb.

Answer: We are very sorry for our mistakes. We have revised the manuscript as suggested.

Question 26：line 208: "literatures" should be "literature"

Answer: We are very sorry for our mistakes. We have revised the manuscript as suggested.

---

## [Decision Letter · Decision Letter 1]

4 Aug 2020

Prognostic effect of pretreatment albumin-to-alkaline phosphatase ratio in human cancers: a meta-analysis

PONE-D-20-05632R1

Dear Dr. Gu,

We’re pleased to inform you that your manuscript has been judged scientifically suitable for publication and will be formally accepted for publication once it meets all outstanding technical requirements.

Kind regards,

Jason Chia-Hsun Hsieh, M.D. Ph.D

Academic Editor

PLOS ONE

Additional Editor Comments (optional):

All the questions were answered adequately.

Reviewers' comments:

Reviewer's Responses to Questions

**Comments to the Author**

1. If the authors have adequately addressed your comments raised in a previous round of review and you feel that this manuscript is now acceptable for publication, you may indicate that here to bypass the “Comments to the Author” section, enter your conflict of interest statement in the “Confidential to Editor” section, and submit your "Accept" recommendation.

Reviewer #1: All comments have been addressed

Reviewer #3: All comments have been addressed

2. Is the manuscript technically sound, and do the data support the conclusions?

Reviewer #1: Yes

Reviewer #3: Yes

3. Has the statistical analysis been performed appropriately and rigorously? 

Reviewer #1: Yes

Reviewer #3: Yes

4. Have the authors made all data underlying the findings in their manuscript fully available?

Reviewer #1: Yes

Reviewer #3: Yes

5. Is the manuscript presented in an intelligible fashion and written in standard English?

Reviewer #1: Yes

Reviewer #3: Yes

6. Review Comments to the Author

Reviewer #1: This current version (#2) of Guo et al is much improved. The authors have largely addressed the comments/questions raised with discussion and the inclusion of new experiments/controls, which made this study stronger and better-suited for publication in PLOS ONE.

Reviewer #3: All comments have been well addressed and I feel that the manuscript is now acceptable for publication.

7. PLOS authors have the option to publish the peer review history of their article (what does this mean?). If published, this will include your full peer review and any attached files.

Reviewer #1: No

Reviewer #3: No

---

## [Editor Report · Acceptance letter]

11 Aug 2020

PONE-D-20-05632R1 

Prognostic effect of pretreatment albumin-to-alkaline phosphatase ratio in human cancers: a meta-analysis 

Dear Dr. Gu:

I'm pleased to inform you that your manuscript has been deemed suitable for publication in PLOS ONE. Congratulations! Your manuscript is now with our production department. 

Kind regards, 

on behalf of

Dr. Jason Chia-Hsun Hsieh 

Academic Editor

PLOS ONE